# Multidimensional Frailty and Lifestyles of Community-Dwelling Older Portuguese Adults

**DOI:** 10.3390/ijerph192214723

**Published:** 2022-11-09

**Authors:** Ana da Conceição Alves Faria, Maria Manuela F. P. S. Martins, Olga Maria Pimenta Lopes Ribeiro, João Miguel Almeida Ventura-Silva, Esmeralda Faria Fonseca, Luciano José Moreira Ferreira, Paulo João Figueiredo Cabral Teles, José Alberto Laredo-Aguilera

**Affiliations:** 1Abel Salazar Biomedical Sciences Institute, University of Porto, Rua Jorge de Viterbo Ferreira 228, 4050-313 Porto, Portugal; 2North Region Health Administration, 4000-447 Porto, Portugal; 3CINTESIS@RISE, 4050-313 Porto, Portugal; 4Nursing School of Porto (ESEP), 4200-072 Porto, Portugal; 5Centro Hospitalar Universitário de São João, 4200-319 Porto, Portugal; 6School of Economics, University of Porto, 4200-465 Porto, Portugal; 7Laboratory of Artificial Intelligence and Decision Support—INESC Porto LA, 4200-465 Porto, Portugal; 8Facultad de Fisioterapia y Enfermería, Campus de Fábrica de Armas, Universidad de Castilla-La Mancha, Av de Carlos III, nº 21, 45071 Toledo, Spain; 9Multidisciplinary Research Group in Care (IMCU), University of Castilla-La Mancha, 45005 Toledo, Spain

**Keywords:** frailty, aged, lifestyle

## Abstract

(1) Background: Lifestyles are referred to as conditioning factors for the frailty of older adults. However, there are few studies that explore its association. The objective of the present study is to analyze the association between sociodemographic, clinical, and lifestyle factors of older adults people with multidimensional frailty. (2) Methods: Descriptive and correlational study carried out with older adults people registered in a Health Unit in Portugal. Data were collected through a sociodemographic and clinical questionnaire and application of the Individual Lifestyle Profile and Tilburg Frailty Index to assess the lifestyles and multidimensional frailty of older adults, respectively. This last instrument, being of a multidimensional nature, assesses not only physical, but also psychological and social frailty, with a cut-off point of 6. (3) Results: Of the 300 older adults who participated, most were female (60.3%) and had a mean age of 81.34 ± 6.75 years. Moreover, 60.3% of the sample were frail older adults. Gender, marital status, number of household members, number of chronic diseases, number of daily medications, self-perception of health status and lifestyle and use of a walking device were associated with multidimensional frailty (*p* ≤ 0.001). Healthy eating habits, physical activity, relational behaviour, preventive behaviour, and stress management were significantly associated with lower physical, psychological, and social frailty (*p* ≤ 0.001). (4) Conclusions: When community health workers are aware of multidimensional frailty predictors and their components, they can intervene early and, consequently, delay the onset and progression of frailty in older adults.

## 1. Introduction

World population ageing [1] has led to the need for the definition and implementation of strategies to promote active and healthy ageing [2]. Though the continued increase in life expectancy is a great achievement, it is a challenge to keep older adults healthy and maintain their quality of life. Even though some people remain relatively healthy and resilient as they age, others become fragile, and as such more vulnerable to external and/or internal stressors, as well as at high risk of adverse events, such as falls, hospitalizations, disability, and institutionalization [1].

Frailty is common among older adults. In Portugal, data concerning the prevalence of frailty among community-dwelling people, in different studies, ranged from 15.6% to 54.8%, varying according to the screening instrument used [3,4,5,6]. This condition deteriorates not only the functional and cognitive capacity of older adults [7], but also the quality of life of older adults themselves and their families, caregivers, and society, hence it represents a public health priority that requires an early intervention [2].

In that regard, in the last decades, research has focused on this issue. There are different instruments for assessing frailty and, among the best-accepted models in the scientific community, we highlight Fried’s phenotypic or biological model [8], the accumulated deficit model [9], and the integral model [7]. While Fried’s biological model [8] and Rockwood’s model of accumulated deficits [9] are one-dimensional models centered on the physical domain, the integral model is a multidimensional model [7]. This model advocates that fragility can no longer be considered a syndrome exclusively focused on the physical domain, being distinct from disability and comorbidity. This model defines frailty as a holistic condition referring to a dynamic state that affects the person who experiences losses in one or more domains of human functioning (physical, psychological, and social) [5,7].

Despite the different definitions and screening methods, there is a consensus that it is important not only to identify frail older adults or those at risk of frailty, but also their determinants, in an attempt to prevent or delay frailty, as it is a dynamic and reversible process [2].

Health professionals in the community play an important role in the promotion of active and healthy ageing and, although nurses cannot intervene in non-modifiable determinants of frailty such as age, gender, ethnicity, marital status, income, and education, among others, they can act on modifiable life course determinants, namely lifestyles, as described in the comprehensive conceptual model of frailty [5,7], through the implementation of active aging programs and projects and the promotion of healthy lifestyles.

Some studies report that unhealthy lifestyles characterized by smoking, excessive alcohol use, poor eating habits, and low physical activity are associated with physical frailty assessed through the frailty phenotype [10,11], while others associate unhealthy lifestyles with multidimensional frailty [12,13]; however, there are still few studies, none known in Portugal, that associate each component of lifestyle with multidimensional frailty.

Hence, our objective was to analyze the association between sociodemographic, clinical factors and the lifestyles of older adults with multidimensional frailty.

## 2. Materials and Methods

This is a quantitative, correlational, cross-sectional study guided by the Strengthening the Reporting of Observational Studies in Epidemiology (STROBE^®^) tool.

### 2.1. Data Collection, Participants, and Procedures

We conducted the study among older adults registered at a Health Care Unit in Northern Portugal, between October 2020 and May 2021, using a non-probability sampling technique.

According to the defined criteria, older adults living at home, aged 65 or older, registered at a Health Care Unit in Northern Portugal, and without cognitive deficits as assessed by applying the Mini Mental State Examination version [14] were included. We excluded all seniors who had total dependence in self-care, using the Barthel Index [15], and impaired communication. To identify and select older adults, the health professionals used the list of those registered at the Health Unit who met the inclusion and exclusion criteria. Subsequently, recruitment took place through telephone contact. Of the 2300 seniors registered, 300 older adults agreed to participate in the study and 40 refused to participate. The sample size obtained was larger than a previous study carried out in Portugal on this topic [10].

After selection, recruitment, and acceptance, we scheduled data collections divided among the main investigator and three nurses previously trained to apply the form. Each form was applied by a nurse.

### 2.2. Instrument

During the data collections, a questionnaire previously prepared by the researchers was applied, containing sociodemographic data (gender, age, education, marital status, and household), clinical data (number of chronic diseases, amount of daily medications, self-perception of health status and lifestyle, and use of mobility and walking aid), the Tilburg Frailty Index (TFI) [5,7], and the Individual Lifestyle Profile (ILP) [16].

The TFI is a questionnaire to assess multidimensional frailty and we divided it into two parts: in the first, we recorded the determinants of frailty, while the second part consisted of 15 questions divided into the three components of frailty: physical, psychological, and social. The physical component includes eight questions related to the patient’s perception of their physical health, unexplained weight loss, difficulty in walking, maintaining balance, hearing problems, vision problems, loss of hand strength, and physical tiredness. In the psychological component, we placed four questions related to memory, symptoms of depression, anxiety, and coping strategies. The social component includes three questions regarding household composition, relationship, and social support. All items are rated between 0 and 1, with the cut-off point for frailty being a score of 6. Higher scores represent greater frailty [5,7].

We applied the ILP to assess the lifestyles of older people. The questionnaire integrates five dimensions, namely, nutrition, physical activity, preventive behaviour, relational behaviour, and stress control. In each dimension, three questions are included and each of the answers are graded from 0 to 3. Answers 0 and 1 indicate health risk behaviour (negative profile) and answers 2 or 3 indicate a positive lifestyle profile. Each dimension can thus range from 0 to 9 points, where up to 3—negative profile; 4 to 6—intermediate (can improve), and 7 to 9—positive profile. The lower the score, the greater the need for behavioural change [16].

### 2.3. Data Analysis

We used the statistical program IBM-SPSS (version 27.0) to analyze the data. The description of the sample was performed using the mean, standard deviation, median, and interquartile range (IQR) according to the results of the Shapiro–Wilk test for quantitative variables and absolute and relative frequencies for qualitative variables. We also performed bivariate analyzes using Mann–Whitney, Kruskal–Wallis, and Spearman correlation tests, as well as multivariate linear regression model with a statistical significance level of *p* ≤ 0.05. We previously checked the normality of the data using the Shapiro–Wilk test [17]. The Cronbach alpha values of the TFI and ILP were 0.727 and 0.858, respectively.

### 2.4. Ethical Considerations

The Ethics Committee and the Board of Directors of the Health Unit where the older adults are registered approved the study, according to Opinion n. º 24/2020. All of the older adults gave their consent and were informed about the research objectives, the procedures adopted, and the guarantee of anonymity and data confidentiality. For the use of the TFI and ILP, we asked for the author’s permission [5,7,16].

## 3. Results

Of the 300 older adults who participated in the study, the majority (60.3%) were female, married (58%), and with a mean age of 81.3 ± 6.7 years. The older adults’ education ranged from illiteracy (10.3%) to a degree (1.7%), with the majority (88%) having a 4-year degree. Two-person households formed the majority (50.7%), with a strong predominance of small families. We highlight the fact that a relevant proportion of the older adults have a one-person household (17%). The majority of the older adults perceived their health status as acceptable (44.7%) and, concerning their lifestyle, they considered that it is neither very healthy nor very unhealthy (45.3%).

Regarding the clinical conditions of the older adults, we found that almost all of the older adults reported having up to five chronic diseases (91.3%) and the average number of chronic diseases among the participants was three. Almost all of the older adults (98.3%) take medications, with the average number being 5.6 medications. Most of the older adults do not use a walking device (71%) and, among the determinants of frailty, we underline the experience of a serious illness in themselves (31.0%), according to Table 1.

We found a predominance of frail older adults, representing 60.3% of the sample. Among the physical components of multidimensional frailty with the highest prevalence, problems in daily living due to impaired vision (82.0%) and hearing (63.7%), difficulty in maintaining their balance (54.0%), difficulty in walking (50%), as well as perceived physical fatigue (66.0%) are highlighted. Regarding the psychological components of multidimensional frailty, we found that most seniors have been feeling down (68.3%) and have been feeling nervous or anxious (57%) during the past month. With regard to the social components of multidimensional frailty, we found that 63.3% of the older adults sometimes miss having people around them and 86.3% report not receiving enough support from other people, according to Table 2.

Regarding the lifestyle of the older adults, we found that the majority have a negative individual lifestyle profile regarding physical activity habits (91.7%) and relational behaviour (76%), according to Table 3.

From the bivariate analysis, we found that marital status, self-perception of health status and lifestyle, number of chronic diseases, amount of medications, and use of mobility and walking aid were associated with multidimensional frailty and its components. Meanwhile, gender was significantly associated with all components except social frailty. In turn, age had a moderate and positive significant correlation with physical frailty, according to Table 4.

The relationship between the components of frailty and the dimensions of the individual lifestyle profile scale were determined using Spearman’s correlation coefficient, and we found significant, negative correlations with all components of multidimensional frailty and with total frailty. The correlation with the social components is somewhat weak and the correlations with the remaining frailty components and with total frailty are moderate, whereas the correlations with the physical components and with total frailty are stronger, as shown in Table 5.

To construct the predictive model of multidimensional frailty associated with lifestyles, we performed a multivariate analysis and, for this purpose, we adjusted a multivariate regression model where the explained variable is frailty, i.e., each component of multidimensional frailty and total frailty and all of the others are the explanatory variables. We selected the remaining explanatory variables by stepwise estimation (*p*-value of 5% and 10% as input and removal criteria, respectively). We also highlight the fact that the correlation matrix between all quantitative independent variables was previously calculated and we detected no problems concerning multicollinearity. All regressions are globally significant (significant F-statistic), with a moderate quality of fit, as determined by the coefficient of determination. Below, we display the explanatory variables retained in the model and their effect on the explained variable in Table 6.

We observed that, among the ILP dimensions, only preventive behaviour was not identified as a predictor of multidimensional frailty and, the more negative the lifestyle among the older adults with respect to nutritional habits, physical activity, relational behaviour, and stress control, the greater the average physical frailty of the older adults. Nutritional habits cross-cuttingly influence physical, psychological, and social frailty in the older adults; relational behaviour influences the physical component and multidimensional frailty and stress control only is not a predictor of social frailty.

The number of illnesses causes physical, psychological, social, and multi-dimensional frailty, with seniors who have had a serious illness (of themselves) in the last year having, on average, less frailty than those who have not. Regarding the experience of a serious illness of a loved one in the last year, we found that the older adults who experienced it have on average more physical, psychological, and multidimensional frailty than those who did not.

Male gender, divorced marital status, and education were predictors of psychological frailty and married marital status and household size of social frailty.

## 4. Discussion

Frailty caused by advancing age is a major public health problem and a challenge for health care professionals [2]. In our study, most of the older adults were frail; hence, it is in line with several published studies, namely, a study conducted in Portugal, where the prevalence of frailty, assessed by TFI, was 54.8% [5]; a study in Poland where the prevalence was 54.6% [18]; and a study in Brazil with 65.25% [19]. Researchers report that the differences are due to the different sociocultural and economic contexts associated with where the older adults people live [5,6,7,8,9,10,11].

In the bivariate analysis, we found that, among the sociodemographic factors significantly associated with frailty, gender, marital status, and the number of household members are highlighted. The higher prevalence of frailty in females is in accordance with a recent study stating that women, especially after menopause, have greater functional decline, loss of muscle mass, osteoporosis, and a higher prevalence of chronic diseases, and hence a higher likelihood of frailty [20].

We also confirmed a higher risk of frailty in single individuals compared with married, widowed, and divorced individuals, as described in a recent systematic review and meta-analysis [21].

Regarding the age factor, we found a significant and moderate correlation with physical frailty (r = 0.615; *p* = 0.029), as described in a Portuguese study [22]. While age may be a risk factor for physical fitness owing to human physiology, age may not necessarily be a specific risk factor for psychological and social frailty, because, in our study, we found a weak correlation. Certain life events, such as the death of a spouse, can negatively influence the psychological and social component of frailty, because the person may sometimes begin to live alone and hence become isolated. The transition due to an adverse life event leads to isolation and not age.

Regarding education, although several studies mention it as a protector for frailty conditions [23], in our research, a significant correlation was only observed with the physical component score (*p* = 0.009). However, as it is very weak (r = −0.151), we concluded that this association is almost non-existent.

Regarding the number of household members, we found that the correlations with the components of physical and psychological frailty are non-significant, thus it is assumed that there is no relationship between these and household size; nevertheless, the correlation with social frailty was significant (*p* < 0.001), being negative and moderate (−0.425). In turn, the correlation with the total frailty score was significant and negative, but, because it was very weak, we concluded that this association was almost non-existent. These data make a whole lot of sense because living alone can lead to social isolation and it contributes to the sense of missing people and of not receiving enough social support, as described in several studies [12,24].

Regarding clinical conditions, a significant and positive association was observed between the number of chronic diseases and multidimensional frailty and all components of frailty, as described in other studies [25,26]. Experiencing a serious illness in the past year also had a significant association with frailty status, ratifying a recent survey in Europe [25]. Previous studies report that chronic diseases are considered the main risk factors for frailty and they are mainly expressed in markers of physical and psychological frailty [9,27]. In turn, only the correlation with the social component of frailty was weak, whereas the other three were moderate, which displays a very relevant association as it demonstrates that chronic diseases not only condition frailty situations in older adults, but also the social context [28].

As for medications, there was an association with all components of frailty, especially with physical and psychological frailty, thus corroborating what we found described in some studies, that is, that aging is associated with increased prevalence of illnesses and increased need for various medications [24,29]. Consequently, therapeutic reconciliation is necessary to assess the real need for the medication given its adverse effects and the increased risk of its inappropriate use [24,29].

Concerning self-perception of health status, we found it to be significantly associated with all components of multidimensional frailty (*p* < 0.001), corroborating previous studies that state that a low perception of health status is a predictor of frailty in older adults [30].

Regarding the lifestyle of the older adults, we found that those who had a healthier lifestyle and, specifically, better nutrition, physical activity, preventive behaviour, relational behaviour, and stress control had lower frailty scores assessed through the TFI. Their perception of lifestyle was also associated with frailty in all components, as well as multidimensional frailty. These results show that the older adults are aware of their condition, which is in agreement with previously conducted studies [6,29].

In accordance with a previous study [30] and in our research, healthy eating habits were found to be associated with lower frailty (*p* < 0.001), especially for the physical, psychological, and multidimensional frailty components, where correlations were moderate and somewhat weak for the social components. Although eating is a social act [31], incorrect eating habits express themselves on the physical and psychological health of older adults [32,33].

Physical activity was the lifestyle factor most strongly associated with frailty, especially physical frailty, which is in line with what is described in several studies stating that moderate to vigorous physical activity as well as physical activity at home are protective factors against frailty in older adults [34,35]. Nevertheless, as people age, sedentary behaviour and isolation increase [36,37], which further contributes to frailty [38]. It is likely that frail people are unable to be physically active owing to real and perceived barriers that represent obstacles to the adoption and maintenance of regular physical activity; however, we should not exclude them from physical activity programs because they are unlikely to adhere and thus it is important to find strategies that motivate them to participate [39]. Several studies demonstrate that multicomponent physical activity and exercise, in addition to delaying frailty are effective interventions to reduce frailty [2].

A recent study indicated that seniors who participated in social or communal activities as sports clubs, group games, or volunteer activities had a lower risk of functional disability [40]. In the present study, decreased relational behaviour, that is, poor establishment of social networks among family, friends, neighbors, or community [16], was also found to associate with frailty, especially physical frailty, which is in accordance to what has been described in several studies that social participation, social support, and social networks are highly related to physical frailty in older adults [24,41,42].

A Chinese study revealed that the social environment plays an important role in the well-being of the older adults and its absence is associated with an increased risk of becoming physically frail [43]. More frequent social practices and occasions for involvement not only help maintain self-identity, but also provide better access to supportive and relational networks and further promote healthier ways of living [44]. Some recent prospective studies have highlighted the relationship between social isolation and frailty by predicting future frailty using social participation [42,45,46].

With regard to stress management, in our study, it was apparent that stress management is inversely associated with multidimensional frailty, particularly psychological, and physical frailty. A recent study reports that high levels of anxiety and/or depression were associated with frailty and a greater likelihood of adverse outcomes [47]. Another study reports that positive attitudes towards aging, such as self-esteem, optimism, and resilience, seem to decrease the risk of frailty [48].

Regarding preventive behaviours, namely, deprivation of addictive habits, self-management and safety, and citizenship habits, despite being associated with frailty, it was found that they were the habits in which the correlation with multidimensional and physical, cognitive, and social frailty was weaker. A recent study reports that smoking is associated with both the onset and worsening of frailty [49], and both smoking and alcohol habits negatively influence not only physical frailty, but also psychological and social frailty in older adults [50]. Concerning the ability to self-manage health, we know that the ability of the older adults to self-regulate or self-manage their health influences the aging processes. In our study, there was a correlation between health self-management and frailty levels among the older adults, which is in agreement with published studies [51,52]. Providing frail older adults with access to educational information and guidance on self-management of health and frailty improves their understanding and knowledge of their condition [53]. Hence, increased knowledge empowers them to actively participate in health and engage in shared health decision-making behaviours [54]. As far as safety habits are concerned, studies show that frail older adults drive less, which demonstrates their responsibility in promoting road safety [55].

The predictive model of multidimensional frailty and its components associated with the lifestyles of older adults was considered satisfactory and we verified that, among the dimensions of the individual lifestyle profile, four (nutrition, physical activity, relational behaviour, and stress control) were predictors of physical frailty, two of psychological frailty, one of social frailty, and three of multidimensional frailty. These results indicate that negative lifestyles manifest themselves more in the physical component than in the other components, because the somatic, physiological, and sensory changes that occur with aging mainly reduce the physical capacity of older adults [8].

Physical activity was only predictive of physical frailty, which is in line with what we described in a systematic review that, despite the different criteria for the detection of frailty, low physical activity is transversal in the different instruments, expressing itself mainly in the functional capacity of older adults [35]. Mobility limitation and sedentary lifestyle during aging associated with loss of strength and/or function cause sarcopenia and physical frailty [56].

As for stress control, it simply is not a cause of social frailty, i.e., resilient and effective stress management translates into less physical and psychological frailty, as we already concluded in the bivariate analysis and described in a recent study [48].

We also found that dietary habits are predictors of all components of frailty as well as multidimensional frailty, i.e., they play a main role in preventing frailty, corroborating recent studies [13,33].

The number of illnesses was also predictive of all components of frailty as well as multidimensional frailty, which is in agreement with Rockwood’s deficit accumulation model [9].

Experiencing a serious illness of a loved one was predictive of physical, psychological, and multidimensional frailty, meeting Gobbens’ multidimensional model [7] and Papathanasiou’s study [25]. Although it has long been known that a healthy lifestyle can prevent chronic diseases, disability, cognitive decline, and early death, healthy aging is not merely the absence of disease, but rather i tis highly influenced by a combination of health domains [57], making it important to independently and simultaneously analyze these factors in health promotion programs [58].

The fact that the nature of this study is cross-sectional and was conducted in the context of only one Health Care Unit does not allow for generalization of the data and strict cause–effect interpretations of the associations between sociodemographic, health conditions, lifestyles, and multidimensional frailty. Future longitudinal studies in different contexts that take into account changes in frailty status over time need to draw stronger conclusions about the effect of different dimensions of the individual lifestyle profile on the aging process and which dimension or which combination of lifestyle dimensions are most important for healthy aging.

## 5. Conclusions

Our study confirms previous findings, namely that an unhealthy lifestyle is an important modifiable factor that can delay multidimensional frailty in older adults. Until now, it was not known how each lifestyle component was associated with multidimensional frailty and its physical, psychological, and social components and which ones had the greatest influence on the frailty condition.

Most studies that simultaneously associated frailty with lifestyles only considered levels of physical activity, diet, and addictive behaviors such as smoking and alcohol habits. In our study, it was also found that relational and stress control behaviors are predictors of multidimensional frailty.

Thus, community nurses, by identifying the predictors of frailty, namely lifestyles (eating habits, physical activity, relational behavior, and stress management), can intervene early and in an individualized way, delaying frailty.

In this manner, when nursing plans are implemented as soon as negative lifestyles are identified, it is possible to improve the quality of life of older adults, reducing adverse events and premature deaths.

## Figures and Tables

**Table 1 ijerph-19-14723-t001:** Socio-demographic and clinical characterization of the participants (N = 300).

Variables	
Gender *n* (%)	
Woman	181 (60.3%)
Men	119 (39.7%)
Age (years) Mean; Standard Deviation	81.3 ± 6.7
Marital status *n* (%)	
Single	13 (4.3%)
Married	174 (58.0%)
Divorced	6 (2.0%)
Widower	107 (35.7%)
Education (years) Mean; Standard Deviation	4 ± 3.0
No of household members	2.4 ± 1.2
Self-perception of health status *n* (%)	
Bad	62 (20.7%)
Acceptable	134 (44.7%)
Good	91 (30.3%)
Very good	12 (4.0%)
Excellent	1 (0.3%)
Self-Perception of Lifestyle *n* (%)	
Unhealthy	30 (10.0%)
Neither too much nor too little healthy	136 (45.3%)
Healthy	134 (44.7%)
No of chronic diseases Median; IQR	3 (2)
No. of daily medications Median; IQR	5 (4)
No. of older adults who use a walking device *n* (%)	87 (29.0%)
Cane	72 (82.8%)
Walker	9 (10.3%)
Wheelchair	6 (6.9%)
Determinants of frailty—Situations experienced in the last year *n* (%)	
Death of a loved one	25 (8.3%)
Serious illness in themselves	93 (31.0%)
Serious illness of a loved one	50 (16.7%)
Divorce	2 (0.7%)
Traffic Accident	3 (1.0%)
Crime	4 (1.3%)

**Table 2 ijerph-19-14723-t002:** Characterization of physical, psychological, social, and total frailty of the entire sample and the frail older adults using the TFI.

Variables		Frail (N = 181)	Total (N = 300)
Components of Multidimensional Frailty	Physical Frailty Median; IQR	6 (2)	5 (4)
	1—Poor physical health *n* (%)	145 (80.1)	155 (51.7)
	2—Unintentional weight loss *n* (%)	19 (10.5)	19 (6.3)
	3—Difficulty in walking *n* (%)	124 (68.5)	150 (50.0)
	4—Difficulty in maintaining balance *n* (%)	145 (80.1)	162 (54.0)
	5—Poor hearing *n* (%)	139 (76.8)	191 (63.7)
	6—Poor vision *n* (%)	158 (87.3)	246 (82.0)
	7—Lack in hand strength *n* (%)	115 (63.5)	119 (39.7)
	8—Physical tiredness *n* (%)	168 (92.8)	198 (66.0)
	Psychological Frailty Median; IQR	3 (1)	2 (3)
	1—Problems with memory *n* (%)	105 (58)	109 (36.3)
	2—Feeling down *n* (%)	173 (95.6)	205 (68.3)
	3—Feeling nervous or anxious *n* (%)	155 (85.6)	171 (57.0)
	4—Unable to cope with problems *n* (%)	146 (81.2)	173 (57.7)
	Social Frailty Median; IQR	1 (1)	1 (1)
	1—Living alone *n* (%)	34 (18.8)	48 (16.0)
	2—Miss having people around *n* (%)	157 (86.7)	190 (63.3)
	3—Not receiving enough support *n* (%)	32 (17.7)	41 (13.7)
Total Frailty Median; IQR		11 (4)	7 (7)

IQR—interquartile range.

**Table 3 ijerph-19-14723-t003:** Classification of the dimensions of the individual lifestyle profile and individual lifestyle (N = 300).

Variables		Classification					
		Negative		Intermediary		Positive	
		*n*	%	*n*	%	*n*	%
Dimensions of the Individual Lifestyle Profile Total	Nutrition	90	30.0	170	56.7	40	13.3
	Physical activity	275	91.7	18	6.0	7	2.3
	Preventive behaviour	34	11.3	113	37.7	153	51.0
	Relational behaviour	228	76.0	53	17.7	19	6.3
	Stress Control	109	36.3	133	44.4	58	19.3
Individual Lifestyle Profile Total		112	37.3	168	56.0	20	6.7

**Table 4 ijerph-19-14723-t004:** Association between socio-demographic and clinical conditions with physical, psychological, social, and total frailty from TFI.

Variables		Components of Multidimensional Frailty									Total Frailty		
		Physical Frailty			Psychological Frailty			Social Frailty					
		$\bar{x}$	Coef.	*p*	$\bar{x}$	Coef.	*p*	$\bar{x}$	Coef.	*p*	$\bar{x}$	Coef.	*p*
Gender													
	Female	4.5	----	0.002 *	2.4	----	0.002 *	0.99	----	0.086 *	7.9	----	0.001 *
	Male	3.6	----		1.8	----		0.84	----		6.3	----	
Age		---	0.615	0.029 **	---	0.205	<0.001 **	---	0.143	0.013 **	---	0.077	0.183 **
Marital Status													
	Single	6	----	<0.001 ***	3.4	----	0.003	1.5	----	<0.001 ***	10.9	----	<0.001 ***
	Married	3.8	----		2	----		0.75	----		6.6	----	
	Divorced	2.3	----		1	----		1.2	----		4.5	----	
	Widower	4.5	----		2.4	----		1.2	----		8	----	
Education		----	0.151	0.009 **	----	0.065	0.264 **	----	0.033	0.955 **	----	0.072	0.217 **
No of household members		----	0.062	0.283 **	----	0.069	0.232 **	----	0.425	<0.001 **	----	0.148	0.010 **
No. of chronic diseases		----	0.664	<0.001 **	----	0.548	<0.001 **	----	0.283	<0.001 **	----	0.638	<0.001 **
No. Medications		----	0.644	<0.001 **	----	0.526	<0.001 **	----	0.285	<0.001 **	----	0.626	<0.001 **
Self-assessment of health status													
	Unhealthy	6	----	<0.001 ***	3.4	----	<0.001 ***	1.4	----	<0.001 ***	10.9	----	<0.001 ***
	Neither too much nor too little	5.5	----		3.1	----		1.2	----		9.8	----	
	Healthy	2.4	----		0.96	----		0.57	----		3.9	----	
Lifestyle self-examination													
	Bad	5.6	----	<0.001 ***	3	----	<0.001 ***	1.4	----	<0.001 ***	10.1	----	<0.001 ***
	Acceptable	4.9	----		2.8	----		1	----		8.8	----	
	Good	2.4	----		0.87	----		0.54	----		3.8	----	
	Very good	1.3	----		0.77	----		0.46	----		2.5	----	
Use of a walking device													
	No	3.6	----	<0.001 *	1.9	----	<0.001 *	0.85	----	0.005	6.3	----	<0.001 *
	Yes	5.5	----		2.8	----		1.1	----		9.5	----	

$\bar{x}$—mean; Coef.—coefficient; * Mann–Whitney; ** Spearman correlation; *** Kruskal–Wallis.

**Table 5 ijerph-19-14723-t005:** Correlations between dimensions of the individual lifestyle profile and physical, psychological, social, and total frailty from the TFI.

Dimensions of the Individual Lifestyle	Components of Multidimensional Frailty						Total Frailty	
	Physical Frailty		Psychological Frailty		Social Frailty			
	Coef.	*p **	Coef.	*p **	Coef.	*p **	Coef.	*p **
Nutrition	−0.407	<0.001	−0.479	<0.001	−0.384	<0.001	−0.486	<0.001
Physical activity	−0.575	<0.001	−0.388	<0.001	−0.188	<0.001	−0.525	<0.001
Preventive Behaviour	−0.371	<0.001	−0.361	<0.001	−0.231	<0.001	−0.400	<0.001
Relational Behaviour	−0.573	<0.001	−0.369	<0.001	−0.184	0.001	−0.510	<0.001
Stress Control	−0.411	<0.001	−0.434	<0.001	−0.223	<0.001	−0.446	<0.001
Individual Lifestyle Total Profile	−0.646	<0.001	−0.574	<0.001	−0.350	<0.001	−0.663	<0.001

Coef.—coefficient; *p* *—Spearman’s correlation.

**Table 6 ijerph-19-14723-t006:** Multivariate linear regression model with frailty components from the TFI as dependent variables.

Components of Multidimensional Frailty	Independent Variables	Estimated Parameters		
		Estimate	*t*	*p*
Physical Frailty	Serious illness of a loved one	0.658	2.803	0.005
	No. of chronic diseases	0.570	9.474	<0.001
	Nutrition	−0.101	−2.908	0.037
	Physical activity	−0.235	−3.582	<0.001
	Relational behaviour	−0.200	−4.040	<0.001
	Stress control	−0.097	−2.388	0.018
	R^2^	0.578		
	Statistics. F	66.5		<0.001
Psychological Frailty	Male gender	−0.290	−2.070	0.039
	Divorced	−1.362	−2.751	0.006
	Education	0.085	3.566	<0.001
	Serious illness of a loved one	0.432	2.397	0.017
	No. of chronic diseases	0.375	8.665	<0.001
	Nutrition	−0.156	−4.302	<0.001
	Stress control	−0.141	−4.631	<0.001
	R^2^	0.462		
	Statistics. F	35.6		<0.001
Social Frailty	Marital status—married	−0.334	−4.241	<0.001
	Household size	−0.215	−6.471	<0.001
	Serious illness of themselves	−0.201	−2.366	0.019
	No. of chronic diseases	0.079	3.297	0.001
	Nutrition	−0.086	−4.369	<0.001
	R^2^	0.312		
	Statistics. F	26.5		<0.001
Total Frailty	No. of chronic diseases	1.034	9.542	<0.001
	Serious illness of a loved one	1.184	2.770	0.006
	Nutrition	−0.387	−4.447	<0.001
	Relational behaviour	−0.306	−3.827	<0.001
	Stress control	−0.270	−3.707	0.003
	R^2^	0.542		
	Statistics. F	69.2		<0.001

## Data Availability

The data that support the findings of this study are available from the corresponding author upon reasonable request.

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
