# Peer review of "Multidimensional Frailty and Lifestyles of Community-Dwelling Older Portuguese Adults"

_ijerph, 2022, doi:10.3390/ijerph192214723_

Round 1

Reviewer 1 Report

The authors present a detailed analysis of frailty to compare sociodemographic and clinical questionnaires with the Tilburg Frailty Index (TFI) and the Individual Lifestyle Profile (ILP).  The results presented in the current manuscript support the evidence found in other studies and sets a starting point to manage frailty and some of its modifiable factors to improve quality of life in the elderly.

Even though the analysis is performed in an extended way, there are several changes to be done in order to improve the understanding of the content and more important some mistaken statistical expressions and lack of information.

Please find my comments in a chronological order, which do not imply an order of importance.

1. Abstract

The authors are required to pay particular attention to preparing their abstract as this is a reflection of their work and may be the only part that is read by some readers. The abstract is already formatted with the required headings: (1) Background, (2) Methods, (3) Results, (4) Conclusions. But the results section of the abstract should summarize the main results supported by actual data in terms of numbers and key characteristics of the study cohort, percentages, percent differences, important correlations, hazard or odds ratios and similar information including also some p-values etc. For clinical studies, please always mention the number of patients and some important characteristics of the study population (e.g. % females, mean age+/-SD, etc.).

Specifically it is difficult to understand that Tilburg Frailty Index (TFI) and multidimensional frailty refer to the same concept. Please extend (2) Methods by specifying the components of TFI and the cut-off value used to define frailty in your population.

2. Material and Methods

- In line 84 it is mentioned that you planned a sample size of 330, for 95% confidence level and a 5% margin of error. A power analysis is usually computed based on a previously known to show superiority of equality. Would you please specify on which study are this calculations based?

- In line 95 you mention for the first time the Individual Lifestyle Profile with an IHLP as abbreviation. Please use ILP, since it is more appropriate. Later the ILP is described in line 107 but with a different abbreviation, i.e. PEVI  for its Portuguese name. Please use always ILP, since it is the English term you used along the text. PEVI is used repeated times wrongly.

- In line 118 you mention “multivariate regression”. The correct term is “multivariate linear regression models” in concordance to Table 5 and since there are several types of regression models. Accordingly, change the title of Table 5, since it is not a logistic regression model what you present in the table.

- Please add in section 2.3 Data Analysis if you present continuous variables as mean±standard deviation (SD) or as median and interquartile range (IQR), depending on your results of Shapiro-Wilk test.

3. Results

- Please add the total population number (n=300) at table 1. It helps the reader to have an overview of the table.

-  I suggest to write “Age (years), mean ± SD” at table 1. Do it accordingly to the other continuous variables and depending on the normal distributions as in the previous comment, e.g. for not normal distributed as “No. of diseases, median (IQR)”.

- The main results of your manuscript compare TFI with other questionnaires. But the results of TFI are written only in the text beginning at line 143. I strongly recommend to add a Table showing all percentages for frail or not frail population, as well as the median values for each component.

- At Table 2, the titles of the columns are wrong (there are two columns as “negative”) and partially in Portuguese. And please add also n=300. Please add at the title: “…and Total Frailty from TFI”.

- At line 175 is written “Table 3. Correlations…”. This should be Table 4. Please add at the end of the title: “… and Total Frailty from TFI”.

- At line 189 is written “Table 4. Multivariate Logistic Regression Model.” It should be “Table 5. Multivariate Linear Regression Model”. I suggest to add at the end: “…Model with Frailty Components from TFI as dependent variables.”

4. Discussion

- In the second paragraph, line 214, you refer to the number of household members. But this result is not included in Table 3. Please add it at the table.

- The same occurs with paragraph at line 233. The results you mention are nowhere to find on the tables.

- In the third paragraph at line 219 you talk about results concerning marital status, saying that they are protective factors for frailty. Nevertheless, the results at table 3 show that Total Frailty is greater than 6 for Single, Married and Widower status, which is a greater value than your cut-off value for Frailty, i.e. >6. Please change this part of the discussion, or specify which aspect might be protective.

As a conclusion, I encourage the authors to improve the manuscript, specially the abstract, so that your research can emphasis the importance of modifiable frailty factors and serve as a pioneer example for further studies.

Author Response

Dear Reviewer:

Enclosed you will find a revision of our manuscript, “Multidimensional frailty and lifestyles of community-dwelling elderly Portuguese people”. We would like to thank you for their thoughtful and constructive comments. We have considered all of the suggestions and have incorporated them into the revised manuscript. Changes to the original manuscript are highlighted in yellow, and we believe that our manuscript is stronger as a result of these modifications. An response to the journal requirements and the reviewers’ comments is presented below.

Reviewer 2 Report

This study collected cross-sectional data about factors that contribute to frailty in the elderly to better understand the relationships between all those factors and levels of frailty. Here are some suggested comments/revisions to help increase the clarity of the paper and better describe how the findings can be useful to health professionals.

-Introduction: you reference three best-accepted models but do not provide any comparative detail about any of them, or explain why you highlighted them. How did those models inform your research design?

- Introduction: Can you describe the difference between frailty and multidimensional frailty? This would help readers understand why you chose your objective in the way you did

-Methods: Did the researcher conduct the interviews with all three nurses together? Or one nurse? Why was a nurse even needed? Please provide clarity on exactly who was present for each interview and what the motivation was

-Methods: The description of the instruments used doesn't include any open-ended interview questions; perhaps the word 'interview' here is misleading? As a reader, I was anticipating more detailed conversation with the participants, not just application/completion of quantitative instruments/surveys

- Table 2: Are the classification variables correct (and all in English?)

- Discussion/Conclusions: The results are presented clearly, and they are well compared to other findings. But, in spite of such a detailed analysis, I don't know how these findings are actionable for health professionals. Other than saying they should look at their patients' lifestyles as that's the piece which is modifiable (which was already known, as you stated in the Introduction), what should people do with these results? What is the next step? 

Author Response

(The authors gave the same response as above.)

Reviewer 3 Report

Thank you for the opportunity to review this manuscript titled “Multidimensional frailty and lifestyles of community-dwelling elderly Portuguese people”. This study investigates the association between socio-demographic, clinical, and lifestyle factors of older Portuguese people with multidimensional frailty, providing support for comprehensive frailty assessment in the community. I believe that the article could be strengthened by including definitions and examples of the key constructs and amending some minor language issues throughout. Please see my specific comments below.

Introduction:

Page 1 – I would suggest using the term “older people” or “older adults” instead of “the elderly” throughout to avoid ageist language.

Page 1 – Given that the concept of conditioning factors is mentioned in the abstract, should there be an explanation of conditioning factors of frailty in the introduction? What theoretical perspective is being drawn upon here?

Page 2 – The authors should consider providing a definition of “multidimensional frailty” and its relevance to the study in the context of other definitions, e.g., physical frailty and social frailty.

Method: 

Page 2 – How was “dependence on self-care” assessed? 

Results:

Table 2 – Please check all table elements are in English language.

Discussion:

Page 9, line 253 – Please correct the following sentence: “corroborates what we found described in several studies”. 

Page 10, 285 – Could the authors define “relational behaviours” and provide examples of the social and communal activities associated with a lower risk of functional disability?

Author Response

(The authors gave the same response as above.)

Round 2

Reviewer 1 Report

Dear authors, thank you for accepting my suggestions and working through the corrections. There are only two minor comments left from my side.

1.     In line 168, there is a typo “%),”  before “according to table 2”.

2.     The list of independent variables at Table 6 are not align, that is, they all start at a different point in the mittle column.It is nicer to read, if they are aligned.

Good look with your future research.

Author Response

Dear Reviewer:

Enclosed you will find a revision of our manuscript, “Multidimensional frailty and lifestyles of community-dwelling elderly Portuguese people”. We would like to thank you for their thoughtful and constructive comments. We have considered all of the suggestions and have incorporated them into the revised manuscript. Changes to the original manuscript are highlighted in yellow, and we believe that our manuscript is stronger as a result of these modifications. An response to the journal requirements and the reviewers’ comments is presented below.

Minor Comments:

  1. In line 168, there is a typo “%),”  before “according to table 2”.

 We appreciate the suggestions and made the correction. We apologized the mistake.

  1. The list of independent variables at Table 6 are not align, that is, they all start at a different point in the mittle column.It is nicer to read, if they are aligned.

 We appreciate the suggestions and made the correction. We apologized the mistake.
